# Adjustable Compression Wraps (ACW) vs. Compression Bandaging (CB) in the Acute Phase of Breast Cancer-Related Arm Lymphedema Management—A Prospective Randomized Study

**DOI:** 10.3390/biology12040534

**Published:** 2023-03-31

**Authors:** Katarzyna Ochalek, Joanna Kurpiewska, Tomasz Gradalski

**Affiliations:** 1Faculty of Motor Rehabilitation, Institute of Clinical Rehabilitation University of Physical Education, 31-571 Krakow, Poland; 2Lymphedema Clinic, St. Lazarus Hospice, 31-380 Krakow, Poland; 3Chair of Orthopaedics, Traumatology and Rehabilitation, Faculty of Medicine and Health Sciences, Andrzej Frycz Modrzewski Krakow University, 30-705 Krakow, Poland; 4Chair of Palliative Medicine, Faculty of Medicine and Health Sciences, Andrzej Frycz Modrzewski Krakow University, 30-705 Krakow, Poland

**Keywords:** breast cancer, lymphedema, compression, multilayer bandaging, compression garments, adjustable compression wraps, self-management

## Abstract

**Simple Summary:**

Compression therapy based on multilayer compression bandaging and compression garment use remains the most efficient component of complex physical therapy (CPT) in lymphedema treatment. The proper self-application of compression bandages is generally considered as problematic. Adjustable compression wraps (ACW) have been proposed as an alternative to the commonly used short-stretch bandages for lymphedema. These systems allow easy application and removal for patients. They have been tested in patients with vascular disorders of the lower limbs and can be considered in breast cancer survivors with upper-limb lymphedema.

**Abstract:**

The objective of this study is to compare the effectiveness, comfort and possibilities of the self-application of adjustable compression wraps (ACW) with compression bandaging (CB) in the acute phase of treatment in advanced upper-limb lymphedema. In total, 36 patients who fulfilled the admission criteria were randomly assigned into ACW-Group (18 patients), or CB-Group (18 patients). Treatment in both groups lasted for two weeks. In the first, all patients were educated in applying adjustable compression wraps (ACW-Group) or self-bandaging (CB-Group) and treated by experienced physiotherapists. In the second week, the use of ACW and CB was continued by the patients themselves at home. In both groups, a clinically significant reduction in the affected limb volume was found after the first week (*p* < 0.001). A further decrease in the affected limb volume within the second week was noted only in the CB-Group (*p* = 0.02). A parallel trend was found in the percentage reduction in the excess volume after one and two weeks of compression therapy. Within two weeks, both groups achieved a significant improvement in decreasing lymphedema-related symptoms, but women from the ACW-Group reported complications related to carrying out compression more frequently (*p* = 0.002). ACW can reduce lymphedema and disease-related symptoms, but based on the results it is difficult to recommend this method as an alternative option in the acute phase of CPT among women with advanced arm lymphedema.

## 1. Introduction

Breast cancer-related lymphedema (BCRL) as a result of lymphatic system insufficiency and impaired lymph transport remains one of major long-term complications among breast cancer survivors [1]. The first stage of BCRL affects only the morphology of the lymphatic vessels and lacks obvious edema, which is subsequently followed by a second stage where tissue swelling and fibrosis occur [2]. Complex physical therapy (CPT), including compression therapy, manual lymph drainage and physical exercise, is widely used and recommended by the International Society of Lymphology (ISL) in lymphedema treatment [3]. Compression therapy based on multi-layer inelastic compression bandaging and compression garment use is still the most efficient component of CPT. 

Despite compression not being curative because it cannot modulate the pathophysiology of lymphedema [4], the positive impact on the lymph pumping function, including a reduction in the pressure in the initial lymphatic vessels, limiting filtration, improving lymphatic reabsorption and stimulating lymphagion contraction as well as the anti-inflammatory activity of the parasympathetic (vagal) system, has been proven in many studies [5,6,7,8]. The most important post-microsurgery lymphaticovenular anastomosis (LVA) is immediate compression to keep anastomosis sites patent, with continuous lymph-to-venous flows via the anastomoses [9]. Immediate compression is also important in liposuction for lymphedema [10,11].

Because lymph is propelled from the extremities to the blood stream not only by intrinsic contraction but also extrinsic forces such as contractions of the skeletal muscles adjacent to the lymphatic vessels, the effectiveness of compression in terms of improving muscle pumping via the veins and lymphatics may also help lymph propulsion by enhancing the extrinsic force. Short-stretch compression characterized with a high level stiffness is more beneficial for the augmentation of muscle pumping than long-stretch compression due to poorer tolerance and the risk of traumatizing soft tissues [12]. In compression treatment for BCRL, both short-stretch bandages and a flat knitted arm sleeve with high stiffness are usually recommended and preferred because they exert a massaging effect in combination with exercise [12]. 

Short-stretch compression bandaging effectively reduces limb volume during the acute, intensive phase of CPT [5,6], while compression garments are mainly used to maintain the achieved effect [13]. In our previous study, we confirmed effectiveness of compression bandaging within the pressure range of 20–30 mmHg in women with advanced arm lymphedema during the intensive phase of treatment [14]. Multi-layer bandaging is also recommended at the maintenance phase, alternatively, with wearing compression sleeves (at night) in more advanced lymphedema [15].

Teaching the self-application of CPT is effective in maintaining or improving its benefits and can be used as a self-care tool in management [16,17,18]. However, the self-application of properly applied multi-layer short-stretch bandages to the upper limbs is generally considered as problematic. Some patients are able to learn the technique and apply the bandage themselves, although this process can be very difficult for the elderly or those with advanced arm lymphedema. Even experienced lymphedema therapists may have difficulty applying bandages correctly [19], especially in more advanced cases of lymphedema. The main problem in multi-layer bandaging is obtaining the optimal pressure under compression. A too-low pressure under the bandage may be ineffective, causing slippage, while on the other hand, excessive pressure may limit the functioning of the upper limb, arterial blood supply and reduce wearing comfort. Therefore, other solutions using inelastic compression therapy are sought.

One option to improve the self-management of patients is innovative adjustable compression wraps (ACW), as an alternative to the commonly used short-stretch bandages [20,21], but used in both the intensive and maintenance phases of CPT. The advantage of this compression device, which is also characterized by high stiffness, is the controlled system of putting on and checking the applied compression force, guaranteeing the effectiveness of self-application in combination with full self-control. They are also easy and less time-consuming on application in comparison to multi-layer bandaging. They may also enhance comfort and lightness, increasing patients’ independence in their daily lives [20,22].

Preliminary clinical trials confirm a similar or even higher effectiveness of ACW in reducing primary lower-limb lymphedema and better comfort at the initial phase of treatment compared to short-stretch bandages [20,23], in mixed-etiology chronic lower-limb edema [24], in chronic venous insufficiency and in the treatment of venous ulcerations [21,25].

In only one randomized clinical trial was the effectiveness of ACW in the treatment of upper-limb lymphedema assessed [26]. There are no studies in which the self-application of adjustable compression products applied by women with upper-limb lymphedema is evaluated.

The objective of this study is to compare the effectiveness, comfort and possibilities regarding the self-application of adjustable compression wraps with multi-layer compression bandaging at the intensive phase of treatment in advanced upper-limb lymphedema among breast cancer survivors.

## 2. Materials and Methods

### 2.1. Participants

During the recruitment period (between January and November 2022), among 65 women after breast cancer treatment admitted to the Lymphedema Clinic, 36 patients fulfilled the admission criteria-stage-II (according to ISL) lymphedema, ≥20% excess limb volumes, positive pitting skin sign, and absent signs of active cancer, venous thrombosis or previous compression. These participants were randomly assigned (simple random allocation by random number generator) into the experimental group with the adjustable compression wrap, ACW-Group (18 patients), or the control with compression bandaging, CB-Group (18 patients). 

The two groups were comparable concerning baseline characteristics, including BMI, type of surgery, additional onco-therapeutic modalities, limb volume and excess volume. They differed only in age. Women in the CB group were older compared to the ACW group (Table 1).

### 2.2. Intervention

In the ACW group, adjustable compression garments (type circaid® juxtafit™ essentials arm sleeve and glove with hand-piece, Medi-Bayreuth) were fitted based on individual limb measurements and they were used for 24 h (Figure 1). The kit included a card with a color scale that allowed users to achieve the appropriate pressure of 20–30 mmHg. The next day, in order to refresh and moisturize the skin, these products were removed for a short period prior to the next therapeutic session. 

In the CB group, multi-layer bandaging of the limb was performed using short-stretch bandages with pads (Rosidal K, Lohmann & Raucher, Poland ). At the beginning of bandaging, a cotton tube stockinet was placed on the arm, and a layer of gauze was applied to the fingers and hand. A layer of foam padding (Lymphset: Lohmann & Raucher) was placed on the hand and wrapped around the arm. Three bandages of different sizes (8, 10, and 12 cm in width) were placed around the limb with the first starting at the hand, the second at the wrist and the third beginning below the elbow (Figure 2).

Sub-bandage pressure was measured via the Kikushime device (Kikgel, Ujazd Poland) on the dorsal part of the forearm, as well as on the arm, to ensure a pressure value within the range of 20–30 mmHg, comparable to the pressure used in the compression system among the ACW Group. Bandages were removed for a short time before the therapeutic session the next day, as in the ACW-Group, and put on the next day. The exercise program, including aerobic exercises of the upper limbs combined with deep breathing, was performed in adjustable compression wraps or bandaging for 15 min in 1 session per day. 

Treatment in both groups lasted 10 days (2 weeks, Monday to Friday). In the first week, all patients were treated by 2 experienced physiotherapists and educated in applying adjustable compression wraps (ACW group) or self-bandaging (CB group). During the 2nd week, the use of ACW and CB was continued by the patients themselves at home. Patients were asked to maintain the adjustable compression wraps or compression bandaging even over the weekend (Saturday and Sunday). After 2 weeks, all patients were fitted with made-to-measure compression sleeves in the range 23–32 mmHg (ccl2, according to German classification).

### 2.3. Measurements

Measurements, including physical functioning, limb volumes and disease-related symptoms, were taken before the intervention, as well as after 1 and 2 weeks of treatment in both groups. Circumferential upper-limb measurements were taken with the arm abducted at 30°, starting at the level of the ulnar styloid, every 4 cm proximal to this point along both limbs. The calculation of limb segment volumes was carried out using a simplified frustum formula (summed truncated cone). The excess volume was the difference between the affected and unaffected limbs expressed in liters. The percentage reduction in the excess volume was obtained as follows: (excess volume before treatment–excess volume after treatment) × 100/excess volume before treatment. To quantify lymphedema, which functions independently of the opposite arm and accounts for fluctuations in patient weight, we additionally used the WAC formula (weight-adjusted volume change calculated as WAC = (*A*_2_*BMI*_1_/*BMI*_2_*A*_1_) − 1, where *A*_1_ and *A*_2_ are arm volumes of the treated arm at baseline and after 2 weeks, and *BMI*_1_ and *BMI*_2_ are Body Mass Indexes at the corresponding time points).

Patients rated their physical functioning, disease-related symptoms, compression comfort and complications of compression on a numerical rating scale (NRS) based on the International Compression Club (ICC) Questionnaire-Part for Patients [27]. Firstly, the patients assessed their physical functioning (sum of ability to move wrist, elbow, shoulder; use a spoon; carry out work; complete household chores; practice sports’ carry out leisure activities; social activities, on the Number Rating Scale (NRS; 0 = not able at all, 10 = completely able) and disease-related symptoms (sum of pain; loss of muscle strength; heaviness; swelling; tight skin; tingling; leakage; NRS: 0 = no complaints, 10 = the highest intensity), first without the tested compression and continued after 1 and 2 weeks using the compression device. Complications of compression related to skin irritation, tender spots. skin damage, itching, warmth, throbbing, cramps, cutting in, slippage, local swelling, bulky feeling, and too-tight feeling (NRS: 0 = not present at all, 10 = very obviously present, sum of points) were assessed after 1 and 2 weeks in both groups. Compression comfort (including: easy donning; easy doffing; feeling immediately after donning; feeling during daytime; putting on clothes on compression; feeling at night; the appearance of the garment (NRS: 0 = not able at all, 10 = completely able, sum of points)) was evaluated after the first and second weeks in both groups. 

### 2.4. Statistics

We summarized the baseline demographics using descriptive statistics and means with standard deviations (SD) or medians with interquartile ranges (IQR) in non-normally (according to Shapiro–Wilk test) distributed ordinal quantitative data. To calculate the differences between the two groups with a non-normal data distribution, the Mann–Whitney test was used. Quantitative non-normally distributed variables in one group evaluated twice were analyzed using the Mann–Whitney test. Data measured within one group more than twice were checked by the Friedman test and Wilcoxon test with Bonferroni post hoc correction. A *p*-value of <0.05 was chosen as the level of statistical significance. Data were analyzed using the R program (version 4.2.2), a language and environment for statistical computing (Vienna, Austria) [28].

### 2.5. Sample Size 

Sample size calculations were performed based on the previous study with an evaluation of adjustable compression wrap effectiveness [26]. A total of 18 individuals in each group were necessary to achieve a power of 80%.

## 3. Results

In both groups, a parallel, clinically significant reduction in affected limb volume was found after 1 week of the intensive phase therapy (0.21 L in ACW-Group and 0.28 L in CB- Group, *p* < 0.001). A further decrease in the affected limb volume within the second week of the treatment was noted only in the CB group. No further improvement was observed in the AWC group after the second week. A statistically significant difference in the affected limb volume (3.36 in ACW and 2.76 in CB, *p* = 0.02) was observed between the groups after 2 weeks of therapy. A parallel trend was observed in the percentage reduction in excess volume (−8.3% in ACW and −12.6% in CB, *p* = 0.12) after 1 week and (−10.4 and −15.2, respectively, *p* = 0,17) after 2 weeks of compression therapy, without statistically significant differences. The percentage reduction in WAC (−7.7% in ACW and −10.0% in CB, *p* = 0.12) was comparable in both groups (Table 2).

The level of physical functioning that was comparable before the treatment (54 NRS points in ACW and 51.5 NRS points in CB, *p* = 0.77) significantly decreased in both groups only after 1 week (29 NRS points and 39 NRS points, respectively, *p* < 0.001).

Within 2 weeks of the intensive phase of treatment, both groups achieved a significant improvement in decreasing lymphoedema-related symptoms, but the change in the ACW group was observed only after the second week (from a median of 23 to 13.4 points; *p* = 0.005), while in the CB group, significant changes were observed both after the first and second weeks (from a median 22 to 10 and 4 points, *p* < 0.001). A better improvement in decreasing disease-related symptoms was noted in the CB group (*p* = 0.001) (Table 3).

Despite no differences being observed for the perceived comfort associated with wearing compression systems between the groups, women from the ACW group felt less comfort during the second week of compression compared to the first one (*p* = 0.049). Moreover, women from the ACW group more frequently reported complications related to compression (Figure 3). All women in the ACW group preferred to prolong the treatment to over 2 weeks, based on wearing an individually fitted compression sleeve in ccl2 instead of continuing to use ACW in the maintenance phase.

## 4. Discussion

Pathological changes in the arm during the development of BCRL reduce the flow of lymph through the lymphatic vasculature. Over time, collecting lymphatic vessels exhibit increased inflammation, increased smooth muscle cell coverage, and dilation. Reduced lymph flow leads to valve regression. Consequently, lymph flow moves in the retrograde direction and fluid is no longer transported out of the tissue, resulting in arm lymphedema and fibrosis [2]. Based on this knowledge, it has been hypothesized that the collecting lymphatic vessels fail to pump due to a chronic exposure to elevated afterload. Lymphatic valve leaflets require constant lymph flow and shear stress signaling to escape cell death. Thus, the valve leaflets may disintegrate in the absence of lymph flow through apoptosis [2]. Inelastic compression can improve lymph pump function and this is why this is the most important component in both phases of CPT in the management of upper-limb lymphedema. A goal for future investigation is to show whether inelastic compression can stimulate the valves in humans. However, there would need to be a way to visualize and image the lymphatic valves in vivo.

There is no doubt that multi-layer, short-stretch bandages with higher stiffness are currently the most well-known and most efficient option at the initial phase of CPT in lymphedema treatment, but the self-application of properly applied compression bandages is generally considered as problematic, but manageable [29]. The main difficulty in multi-layer bandaging is to obtain the optimal pressure under compression, not limiting function or reducing the wearing comfort of the upper limb, especially in the elderly or patients with advanced lymphedema.

Adjustable compression wraps have been proposed as an alternative to the commonly used bandages for lymphedema [20,21] and they can be also used during the maintenance phase of CPT. These systems allow easy application and removal for patients. Most studies related to adjustable compression wrap use are concentrated on vascular disorders of the lower limbs. ACWs have been tested by patients with venous insufficiency, only a few in patients with lower extremity lymphedema, and have proven to be effective and well-tolerated [30,31]. There are no studies in which the self-application of adjustable compression products by women with upper-limb lymphedema is evaluated.

To the best our knowledge, this study is the first in which the authors assess the comparative efficacy of the adjustable compression wraps with conventional short-stretch bandages in women with advanced arm lymphedema, performed by a professional physiotherapist during the first week of intensive phase and self-management at home during the second week. 

Only one randomized trial has confirmed the effectiveness of adjustable compression wraps on the upper limbs [26], but the treatment was performed by a professional therapists with the addition of manual lymphatic drainage (MLD). In that study, the effectiveness in terms of the percentage of excess volume reduction and relieving lymphedema-related symptoms in women after breast cancer was comparable to the results obtained in our study; however, the treatment was not based solely on the compression therapy, as additional MLD was not performed. In the present work, no other components of CPT apart from physical exercise were used, and a significant reduction in affected arm volume with improvement in decreasing disease-related symptoms in both groups was noted after applying compression alone. There were no differences between groups in percentage excess volume reduction. Our observations indicate that pressure within the range of 20–30 mmHg in both compression system devices was optimal; they were not only sufficient to obtain positive, clinically relevant results, but also sufficiently safe to ensure good patient compliance. Apart from the pressure, the stiffness of compression is crucial to predict their effectiveness and tolerability for patients [32]. Numerous researchers have demonstrated that the higher the stiffness of a compression, the larger the improvement of hemodynamic parameters and the bigger the reduction in edema [12,32]. Higher stiffness is called inelastic stiffness – with the lower resting pressure and stronger one during physical activity. Comparing stiffness in different compression systems, in elastic or long-stretch compression with lower stiffness, very high pressures of elastic textiles are poorly tolerated and compromise blood flow due to the inward and sustained, forceful recoil of the elastic material [32], which is why this is not recommended in advanced lymphedema. However, the effectiveness of both compression system devices in reducing affected arm volume in our study was noticed mainly in the first week of treatment. No further improvement was observed in the ACW group compared to CB within the second week, despite comparable compliance. Compression bandaging turned out to be more effective, not only in reducing affected limb volume but also in reducing disease-related symptoms during the second week of treatment. Interestingly, the majority of patients in our study were able to learn the technique and applied the bandage themselves for 2 weeks, despite this process even being potentially problematic for some physiotherapists [19].

Controversial information on comfort and tolerance in relation to compression devices comes from our observations, despite no significant differences being noted in physical functioning and comfort within both groups. The ACW group reported complications related to the use of compression more frequently. Women from the ACW group mainly complained about: slippage, local hand swelling, cutting in, too-tight feeling and skin irritation. This contrasts with the results achieved in other studies, in which adjustable compression wraps were considered more comfortable in comparison to bandaging [20,23].

According to the literature on this subject, ACW ought to maintain comfort, but our results indicate that this was difficult to guarantee. Treatment tolerance expressed by the reported complications was surprisingly better in the bandaging group. One of the reasons for worse treatment tolerance in the ACW-G could partially be explained by the tendency towards greater initial excess volume in this group, which was definitely of clinical significance [33]. Therapeutic management in more advanced and chronic cases of lymphedema seems to be more difficult, particularly in hand edema cases. 

In the study by Pujol-Blaya et al. [26], the authors assessed ACW in less severe disease cases (excess limb volume of ≥10%). The second reason for more complications observed within the ACW group could be related the lesser experience of physiotherapists in these new therapeutic systems.

The advantage of ACW is its ease and less time-consuming application in comparison to multi-layer bandaging, which increases independence in everyday activity, but the results from the second week in our clinical trial did not confirm this observation. It turns out that putting on the compression wrap and self-adjusting the appropriate pressure in the upper limb with one hand was not as easy as for the lower limbs. Due to the lack of similar studies evaluating self-management in a homogeneous group of patients as those in our study, a reliable comparison of the obtained results in our study, is difficult. More well-designed and controlled trials are still necessary.

The limitation of this prospective randomized trial is the relatively small number of patients who fulfilled the inclusion criteria. The strength of this study is the selection of a homogeneous group of patients with advanced arm lymphedema (at least 20% excess volume and chronic swelling). The level of compliance regarding the application of compression over 2 weeks in both groups was high.

## 5. Conclusions

ACW, as a part of CPT, can reduce lymphedema and disease-related symptoms but based on the results it is difficult to recommend this method as an alternative option in the acute phase of CPT among women with advanced arm lymphedema. Rigorously designed high-quality randomized trials with larger sample sizes are needed to verify the effectiveness of ACW in BCRL. This management with ACW could be more effective when performed by an experienced physiotherapist. Education on self-management related to adjustable compression wrap application and compression bandaging is still needed. 

## Figures and Tables

**Figure 1 biology-12-00534-f001:**
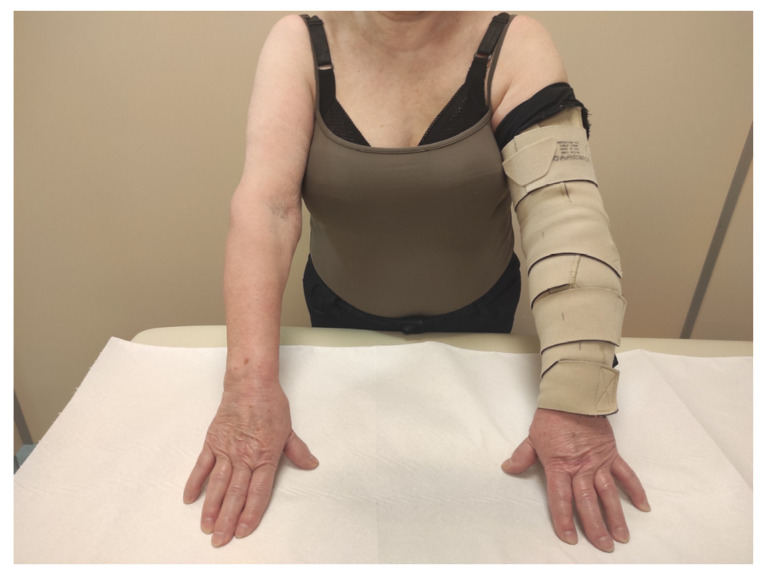
Patient with adjustable compression wrap.

**Figure 2 biology-12-00534-f002:**
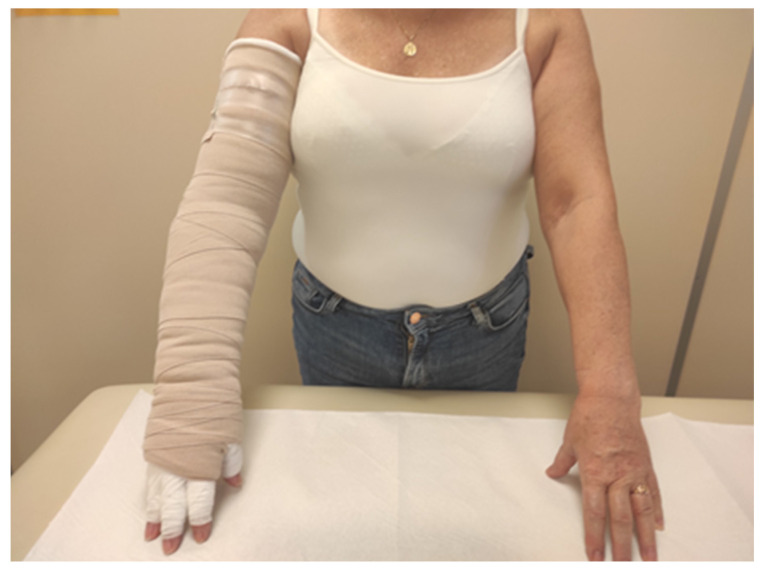
Patient with compression bandages.

**Figure 3 biology-12-00534-f003:**
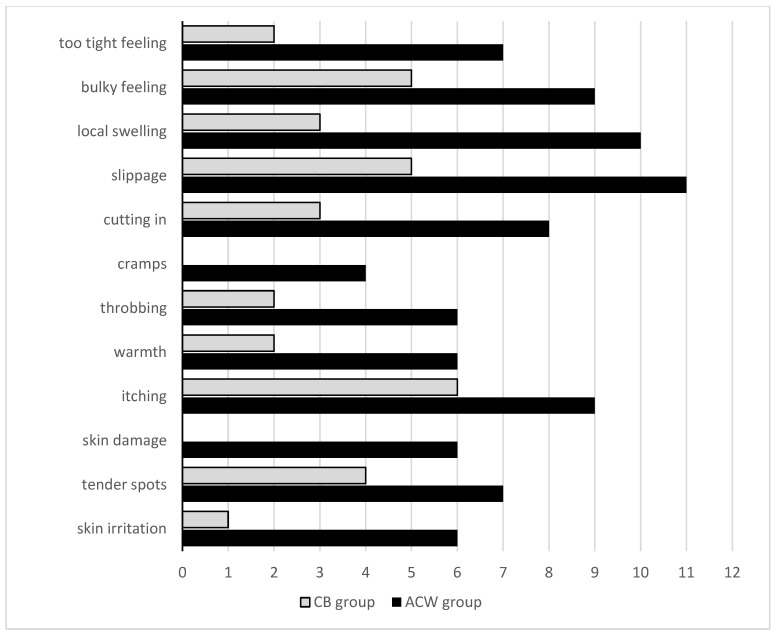
Number of patients with complications of compression within the groups.

**Table 1 biology-12-00534-t001:** Patients characteristics.

Parameter	ACW Group (*n* = 18)	CB Group (*n* = 18)	*p*
Age (years)	62.3 (9.4)	69.6 (9.5)	0.03 *
Body mass index (kg/m^2^)	32.9 (4.1)	28.2 (5.6)	0.67 *
Dominant limb affected (patients no)	10	12	
Months since surgery	55.0 (27.5–122.0)	30.5 (14.0–126.3)	0.45 **
Type of surgery (patients no)			
breast conserving	7	3	
mastectomy	11	15	0.48 ***
axillary nodes dissection	18	18	
Adjuvant therapy (patients no)			
chemotherapy	18	13	
axillary irradiation	16	14	
immunotherapy	4	1	
Affected limb volume (L)	3.57 (3.08–4.22)	3.05 (2.80–3.49)	0.08 **
Opposite limb volume (L)	2.57 (2.19–3.18)	2.3 (2.14–2.50)	0.14 **
Excess volume (%)	39.3 (27.6–51.9)	33.3 (25.3–39.7)	0.41 **

Mean values (standard deviations) or medians (interquartile ranges); * *t*-test, ** Mann–Whitney test, *** McNemar’s test. ACW: adjusted compression wraps; CB: compression bandaging.

**Table 2 biology-12-00534-t002:** Changes in volume within the groups.

Volume	ACW Group (Median, IQR)	CB Group (Median, IQR)	*p*
Affected limb volume (L)			
A before therapy	3.57 (3.08–4.22)	3.05 (2.80–3.49)	0.08 **
B after 1 week	3.36 (2.84–3.96)	2.77 (2.56–3.17)	0.08 **
C after 2 weeks	3.36 (2.84–3.85)	2.76 (2.50–3.12)	0.02 **
	A > B,C	A > B> C	
	*p* < 0.001 *	*p* < 0.001 *	
Excess volume reduction (%)			
after 1 week	−8.3 (−14.38–−5.23)	−12.6 (−14.6–−10.9)	0.12 **
after 2 weeks	−10.4 (−15.4–−5.05)	−15.2 (−17.8–−10.6)	0.17 **
	*p* = 0.42 ***	*p* = 0.3 ***	
WAC (%)	−7.7 (−8.5–−5.1)	−10.1 (−12.3–−6.36)	0.12 **

IQR interquartile ranges; WAC = (*A*_2_*BMI*_1_/*BMI*_2_*A*_1_) − 1, where *A*_1_ and *A*_2_ are arm volumes of the treated arm at baseline and after 2 weeks, and *BMI*_1_ and *BMI*_2_ are Body Mass Indexes at the corresponding time points; * Friedman test (Wilcoxon with Bonferoni post hoc correction); ** Mann–Whitney test; *** Wilcoxon signed rank test.

**Table 3 biology-12-00534-t003:** Patients’ functioning, symptoms and complications of compression.

Domain (NRS)	ACW Group (Median, IQR)	CB Group (Median, IQR)	*p* **
Physical functioning in relation to compression ^a^			
A before therapy	54 (48.0–65.3)	51.5(50.0–61.0)	0.77
B after 1 week	29 (22.3–41.8)	39.0 (24.8–46.5)	0.84
C after 2 weeks	25 (16.0–32.0)	25.5 (15.5–34.0)	0.69
	A > B,C	A > B,C	
	*p* < 0.001 *	*p* < 0.001 *	
Disease-related symptoms ^b^			
before therapy	23.0 (14.3–36.0)	22.0 (17.3–27.3)	0.49
after 1 week	15.0 (12–21.5)	10.0 (3.3–13.8)	0.02
after 2 weeks	13.5 (8.0–27.8)	4.0 (0.3–10.8	0.001
	A > C	A > B> C	
	*p* = 0.005 *	*p* < 0.001 *	
Compression and comfort ^c^			
after 1 week	27.0 (22.5–29.8)	26.0 (19.5–39.0)	0.73
after 2 weeks	37.5 (22.5–43.5)	31.5 (26.5–38.5)	0.40
	*p* = 0.049 ***	*p* = 0.53 ***	
Complications of compression ^d^			
after 1 week	25.5 (18.0–35.5)	7.0 (4.0–14.8)	<0.001
after 2 weeks	23.5 (14.3–40.3)	7.0 (3.3–16.8)	0.002
	*p* = 0.98 ***	*p* = 0.91 ***	

IQR interquartile ranges; NRS Number Rating Scale (0 = minimum/best, 10 = maximum/worst). ^a^ Able to move wrist; elbow; shoulder; use spoon; carry out job; complete household chores; practice sports, carry out leisure activities; social activities (sum of points in NRS; range 0–90). ^b^ Pain; loss of muscle strength; heaviness; swelling; tight skin; tingling; leakage (sum of points in NRS; range 0–70). ^c^ Easy donning; easy doffing; feeling immediately after donning; feeling during daytime; putting on clothes on compression; feeling at night; the appearance of the garment (sum of points in NRS; range 0–70). ^d^ Skin irritation; tender spots; skin damage; itching; warmth; throbbing; cramps; cutting in; slippage; local swelling; bulky feeling; too tight feeling (sum of points in NRS; range 0–120). * Friedman test (Wilcoxon with Bonferoni post hoc correction); ** Mann–Whitney test; *** Wilcoxon signed rank test.

## Data Availability

The datasets used and/or analyzed during the current study are available from the corresponding author on reasonable request.

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
