# Peer review of "Adjustable Compression Wraps (ACW) vs. Compression Bandaging (CB) in the Acute Phase of Breast Cancer-Related Arm Lymphedema Management—A Prospective Randomized Study"

_biology, 2023, doi:10.3390/biology12040534_

Round 1

Reviewer 1 Report

It's an interesting study.

In the abstract: note the difference for the localized complications: complications of compression (p=0.002) and corrected in the BOX (NS), it's a mistake... It's important to not hide the complications or problems with the wraps with the p-value ! And mention with p-value, the more efficacy of the bandaging after 2 weeks.

The last phrase: ACW can reduce lymphedema and disease-related symptoms similarly to multilayer bandages (p=0.001 ?) and can be considered as alternative option in women with advanced arm lymphedema.

Modify : ACW can reduce lymphedema and disease-related symptoms  and could be considered as alternative option in women with advanced arm lymphedema despite more localized complications and less effect after one week in comparison with low-stretch bandages. 

In the Results (table 2): it would be interesting to compare not only % of lymphedema reduction but the % of excess volume after treatment in the 2 groups, maybe it's different.

Protocol in the Ref 13 is published and article has to be cited and analyzed. It's on the same theme (McNeely ML et al. Nighttime compression supports improved self-management of breast cancer-related lymphedema: A multicenter randomized controlled trial. Cancer 2022;128(3):587-59)

Pression compression ccl2 shoud be expressed in mmHg (different from country to another).

Author Response

Thank to the Reviewer for the interest in our work and for helpful comments that will greatly improve the manuscript

Reviewer 2 Report

The aim of study is interesting. The method is reasonable, and not bad about accuracy. The analysis is good. But the conclusion is not supported by the results.

The results shows no advantages of ACW, less comfortable and less effectiveness. If the authors would like to conclude that ACW can be alternative option, some date which show the advantages of AWC must be presented, for example, stableness of wearing, less time consuming. If the authors have no data which shows the advantages of ACW, the authors have no choice but to conclude that ACW has no advantage for upper limb lymphedema, and it can not be recommended.

Abstract

The reviewer thinks figure is not used in abstract.

Introduction

Line 49-50; paragraph change is not necessary in this part.

Line 52-54; Please add the following passage.

Immediate compression is also important in liposuction for lymphedema (Effect of Postoperative Compression Therapy on the Success of Liposuction in Patients with Advanced Lower Limb Lymphedema. J Clin Med. 2021 Oct 22;10(21):4852.), (Comparison of the Effectiveness of Liposuction for Lower Limb versus Upper Limb Lymphedema. J Clin Med. 2023 Feb 21;12(5):1727.)

Material and Methods

Line 108-115; The reviewer hopes the authors to show the clinical photos of each compression garments applied to patients.

Author Response

We thank the Reviewer for the interest in our work and for helpful comments that will greatly improve the manuscript

Round 2

Reviewer 2 Report

The reviewer is satisfied with the revision, and recommends it for publication.